# Novel Activity of ODZ10117, a STAT3 Inhibitor, for Regulation of NLRP3 Inflammasome Activation

**DOI:** 10.3390/ijms24076079

**Published:** 2023-03-23

**Authors:** Ju-Hui Kang, Se-Bin Lee, Jiu Seok, Dong-Hyuk Kim, Gaeun Ma, Jooho Park, Ae Jin Jeong, Sang-Kyu Ye, Tae-Bong Kang

**Affiliations:** 1BK21 Project Team, Department of Applied Life Science, Graduate School, Konkuk University, Chungju 27478, Republic of Korea; 2Department of Pharmacology and Biomedical Sciences, Seoul National University College of Medicine, Seoul 03080, Republic of Korea; 3Department of Biotechnology, Research Institute of Inflammatory Diseases, Research Institute (RIBHS), College of Biomedical and Health Science, Konkuk University, Chungju 27478, Republic of Korea

**Keywords:** ODZ10117, NLRP3 inflammasome, MSU-induced peritonitis, LPS-induced sepsis

## Abstract

The NLRP3 inflammasome serves as a host defense mechanism against various pathogens, but there is growing evidence linking its activation in sterile condition to diverse inflammatory diseases. Therefore, the identification of specific inhibitors that target NLRP3 inflammasome activation is meaningful and important for novel therapies for NLRP3 inflammasome-associated diseases. In this study, we identified a chemical compound, namely ODZ10117 (ODZ), that showed NLRP3 inflammasome-targeting anti-inflammatory effects during the screening of a chemical library for anti-inflammatory activity. Although ODZ was initially discovered as a STAT3 inhibitor, here we found it also has inhibitory activity on NLRP3 inflammasome activation. ODZ inhibited the cleavage of caspase-1 and IL-1β-induced canonical NLRP3 inflammasome triggers, but had no effect on those induced by AIM2 or NLRC4 triggers. Mechanistically, ODZ impairs NLRP3 inflammasome activation through the inhibition of NLRP3–NEK7 interaction that is required for inflammasome formation. Moreover, the results obtained from the in silico docking experiment suggested that ODZ targets NLRP3 protein, which provides evidence for the specificity of ODZ to the NLRP3 inflammasome. Furthermore, ODZ administration significantly reduced MSU-induced IL-1β release and the mortality rate of mice with LPS-induced sepsis. Collectively, these results demonstrate a novel effect of ODZ10117 in regulating NLRP3 inflammasome activation both in vitro and in vivo, making it a promising candidate for the treatment of NLRP3-inflammasome-associated immune disorders and cancer.

## 1. Introduction

Inflammasomes are multiprotein complexes that form with an inflammatory cascade upon exposure to pathogen-associated molecular patterns (PAMPs), such as lipopolysaccharide, bacterial nucleic acid and components, and damage-associated molecular patterns (DAMPs), including ATP, endogenous stress, and uric acid crystals [1]. Four types of pattern recognition receptor (PRR) families recognize PAMPs or DAMPs; among them, NLRP1, NLPR3, NLRC4, and AIM2 form inflammasomes [2]. Once PRRs recognize PAMPs or DAMPs, the adaptor molecule ASC binds to PRRs via each PYD domain and forms an oligomer and ASC speck. Then, the effector molecule caspase-1 (p45) binds to ASC via each CARD domain, forming a complex called inflammasome [3]. As a result of the complex formation, bio-inactive caspase-1 (p45) auto-cleaves itself and forms bio-active caspase-1 (p20), which cleaves the pro-inflammatory cytokine IL-1β and the pyroptosis-mediated protein GSDMD; therefore, an inflammasome is called a caspase-1-activating platform [4].

The NLRP3 inflammasome is the most established inflammasome that responds to endogenous sterile stimuli, such as ATP [5] and metabolic crystals [6,7], as well as external trigger stimuli, such as pore-forming toxins [8] and particulate matter [9]. These stimuli commonly induce potassium efflux, which is an essential NLRP3 inflammasome process [10]. NEK7 is a newly revealed NLRP3 inflammasome regulator that controls downstream of potassium efflux, such as NLRP3 structural changes and ASC oligomers [11,12]. The NLRP3 inflammasome is closely linked to various metabolic diseases, such as type 2 diabetes [13], gout disease [7], atherosclerosis [14], and inflammatory bowel disease [15]; it is also related to neurodegenerative diseases, such as Parkinson’s disease [16] and Alzheimer’s disease [17]. Therefore, many researchers have been making efforts to discover a new inhibitor of NLRP3 inflammasome from compound libraries [18,19] or natural products [20]. In this study, we attempted to screen libraries of compounds that have been approved as drugs or reported for drug development and found that ODZ can inhibit the NLRP3 inflammasome activation.

ODZ is a novel anticancer drug that binds to the SH2 domain of STAT3 and inhibits migration, invasion, and tumor growth in breast cancer and glioblastoma [21,22]. The impact of ODZ on inflammasome activation has not been acknowledged to date; our study revealed its novel activity in inhibiting inflammasome activation, and provides insight into its action mechanism and potential application to an in vivo animal model.

## 2. Results

### 2.1. ODZ Inhibits IL-1β Release and Pyroptosis Induced by Various NLRP3 Inflammasome Triggers

We carried out a screening to identify a compound that can inhibit inflammasome activation. The screening involved testing 3110 compounds, including approved and investigational drugs supplied by the Korea Chemical Bank and academic organizations in the Republic of Korea. The screening was based on the inhibitory effect of compounds on nigericin-induced IL-1β release of THP-1 cells, and we initially identified three compounds that showed an ED50 value of less than 20 μM, and one of them was ODZ (Figure 1A), which has previously been reported as a STAT3 inhibitor. Since the NLRP3 inflammasome is activated by various PAMPs and DAMPs, such as pathogen infection and its components, bacterial toxin, ATP, silica crystals, etc. [23], we further examined if ODZ could inhibit the release of IL-1β from primary mouse macrophages triggered with various stimuli.

To rule out the inhibitory activity being attributed to its cytotoxicity on target cells, an MTT assay was performed in the target cells. As shown in Figure 1B, ODZ did not show significant cytotoxicity up to 40 μM on the mouse macrophages and it suppressed the release of both IL-1β and pyroptosis induced by ATP, nigericin, silica crystals, and imiquimod in a dose-dependent manner (Figure 1C–F).

### 2.2. ODZ Inhibits the Activation of Caspase-1 but Does Not Affect the Expression Level of NLRP3 Inflammasome Components

The cleavage and release of IL-1β are mediated by the activated caspase-1 in an inflammasome complex composed of several proteins, such as NLRP3, caspase-1, ASC, and pro-IL-1β [4]. Therefore, both the activation of caspase-1 and the expression of inflammasome components can affect the release of IL-1β. To investigate by which ODZ inhibits IL-1β release, we examined the impact of ODZ on the cleavage of caspase-1 and its target protein, Gasdermin-D, and the protein level of NLRP3-inflammasome components. The results showed that ODZ inhibited the cleavage of caspase-1, GSDMD, and pro-IL-1β as expected (Figure 2 and Appendix A), but it did not affect the expression of inflammasome-related proteins, such as NLRP3, ASC, GSDMD, pro-caspase-1, and pro-IL-1β. These results indicated that the inhibitory effect of ODZ on inflammasome activation was attributed to the suppression of caspase-1 activation, not to transcriptional regulation.

### 2.3. ODZ Does Not Inhibit AIM2 or NLRC4-Mediated Inflammasome Activation

To determine whether ODZ also inhibits other types of inflammasomes, we examined the impact of ODZ on AIM2- and NLRC4-inflammasome activation. AIM2 is a known sensor of cytosolic dsDNA [24], and its activation can be induced by applying poly(dA:dT), a synthetic B-DNA analog, and NLRC4 inflammasome can be activated by cytosolic bacterial flagellin [25]. The study found that unlike NLRP3 inflammasome activation, ODZ did not inhibit the release of IL-1β and pyroptosis (Figure 3A,B), as well as cleavage of caspase-1 and GSDMD (Figure 3C,D and Appendix A) caused by poly(dA:dT) or flagellin in LPS-primed BMDMs. These results suggest that ODZ has a specific effect on the NLRP3 inflammasome activation, while leaving other type of inflammasomes, such as AIM2 and NLRC4, unaffected.

### 2.4. ODZ Inhibits ASC Speck Formation

Since ODZ inhibits caspase-1 activation in Figure 2, we further investigated its impact on the upstream processes of caspase-1 activation to understand its action mechanism. In the upstream processes of caspase-1, ASC translocates to the perinuclear space and forms ASC specks with oligomerization [3]. Therefore, we examined the effect of ODZ on ASC translocation, its oligomerization, and speck formation induced by NLRP3 inflammasome triggers. The results showed that the translocation of ASC to a Triton X-100 insoluble fraction was suppressed by ODZ in a dose-dependent manner (Figure 4A and Appendix A). Additionally, ODZ inhibited ASC oligomerization and speck formation induced by all NLRP3 inducers used here (Figure 4B–D). Consistent with the previous data in Figure 3A–D, ODZ did not affect the ASC translocation and speck formation in the inflammasome activation induced by poly(dA:dT) and flagellin (Figure 4E–G and Appendix A). Collectively, ODZ suppressed ASC oligomerization, speck formation, and translocation to a detergent insoluble fraction. This result indicated that it might act on the upstream of ASC activation.

### 2.5. ODZ Inhibits the Interaction of NLRP3 and NEK7

Since ODZ inhibits the activation of the NLRP3—but not the AIM2—or NLRC4-inflammasomes, we hypothesized that ODZ might impair NLRP3-specific events, such as NEK7 and NLRP3 interactions [11]. NEK7 is a regulator that regulates NLRP3 oligomerization and activation by interacting with NLRP3 at the downstream level of potassium efflux. To examine the possibility that ODZ interfered with the interaction of NLRP3 and NEK7, we overexpressed NEK7 and NLRP3 proteins in HEK293T to induce their spontaneous interaction with or without ODZ [26]. The results showed that the interaction of NEK7 and NLRP3 was confirmed by co-immunoprecipitation, but their binding was interrupted in the presence of ODZ (Figure 5 and Appendix A). These results suggested that ODZ inhibits NLRP3 inflammasome activation by impeding the interaction between NEK7 and NLRP3.

### 2.6. ODZ Targets the NLRP3

Based on the data presented above, which suggest that ODZ interferes with the interaction between NLRP3 and NEK7, we hypothesized that ODZ might directly bind to either NLRP3 or NEK7. To examine this hypothesis, we used the drug affinity responsive target stability (DARTS) assay, which is a method of identifying potential binding between target proteins and a small molecule. This assay works on the principle that the binding of ligand to the target protein can protect the protein from degradation by protease [27]. To perform the DARTS assay, we incubated ODZ with LPS-primed J774A.1 cell lysates, and then treated with pronase, a mixture of endo- and exoproteases. Both NLRP3 and NEK7 were degraded by pronase treatment, but the proteolysis of NLRP3 protein was reduced by ODZ treatment in a dose-dependent manner, showing similar inhibitory activity to a known NLRP3 inhibitor, MCC950 [28]. In contrast, the degradation of NEK7 or caspase-1 was not affected by either ODZ or MCC950. The degradation of STAT3 was also inhibited by the treatment of ODZ (Figure 6 and Appendix A). Together, these results indicate that ODZ might inhibit NEK7 and NLRP3 interaction through binding with NLRP3.

### 2.7. In Silico Molecular Binding of ODZ with NLRP3 Protein

To assess if ODZ interacted with NLRP3, it was docked into the ADP binding site of the NACHT domain (PDB:7ALV) of NLRP3 using Autodock Vina. The electrostatics of the bound ODZ (−7.7 kcal/mol) in the NLRP3 NACHT domain showed the clear pi–pi interaction with HIS522 and multiple hydrophobic interactions with ARG167, LEU171, ILE234, TYR381, and PRO412 (Figure 7). Furthermore, conventional hydrogen bond interactions involving the residue GLY231 were also observed in docking analysis. Considering the computer simulation results, ODZ could form an ATP/ADP binding pocket in the NACHT domain of NLRP3, thereby potentially inhibiting the ATPase activity of NLRP3.

### 2.8. ODZ Attenuates IL-1β Release in the MSU-Induced Peritonitis Model and Mortality in the LPS-Induced Sepsis Model in Mice

MSU crystals that act as an endogenous danger signal are known to induce gout disease or peritonitis by triggering NLRP3 inflammasome activation [7]. Thereby, to assess the biological significance of ODZ in vitro, the effect of ODZ on IL-1β release into the peritoneal cavity of mice injected with MSU was examined. MSU crystal injection significantly increased IL-1β release, but this increase was suppressed in a dose-dependent manner by ODZ treatment, comparable to MCC950 (Figure 8A).

To validate the biological efficacy of ODZ, it was also tested in an LPS-induced sepsis model. LPS injection in mice leads to systemic inflammation and death, and activation of the NLRP3 inflammasome has been found to contribute to this process [29]. As predicted, pre-treatment with ODZ led to suppression of IL-1β release in a dose-dependent manner (Figure 8B) and significant improvement in mortality rates (Figure 8C), indicating that ODZ has inhibitory activity on NLRP3 inflammasome activation in mice.

## 3. Discussion

An inflammasome is a cytosolic multiprotein complex formed by PRR, pro-caspase-1, and ASC. Although many inflammasomes have been discovered to date, the NLRP3-mediated inflammasome has been well-studied because it senses a broad range of stimuli associated with various inflammatory diseases, including gout, diabetes, and rheumatoid arthritis [7,13,30]. Therefore, the screening and identification of NLRP3 inhibitors has been studied for decades.

In this study, we screened a chemical library derived from various sources and found that a STAT3 inhibitor named ODZ showed a strong inhibitory effect on NLRP3 inflammasome activation. It suppressed the release of IL-1β and caspase-1 from macrophages upon triggering with various NLRP3 agonists, such as ATP, nigericin, and silica. However, ODZ did not affect inflammasome activation induced by poly(dA:dT) or flagellin, indicating that it is specific to NLRP3 inflammasome activation.

Therefore, we explored the mechanism by which ODZ inhibited NLRP3 inflammasome activation. The common events required for the activation of the NLRP3 inflammasome are potassium efflux, NLRP3–NEK7 interaction, ASC speck formation, caspase-1 activation, and bioactive IL-1β release [31]. The most common NLRP3 stimuli, such as ATP, nigericin, and crystals, were dependent on potassium efflux to induce inflammasome activation [10], but potassium efflux was not involved in imiquimod-induced NLRP3 inflammasome activation [32]. ODZ does not seem to inhibit potassium efflux, because it has an inhibitory effect on the imiquimod-induced IL-1β release. However, ODZ did not inhibit downstream pathways, including ASC oligomerization, speck formation, and caspase-1 activation.

Nevertheless, ASC or caspase-1 is an unlikely ODZ target because ASC speck and caspase-1 are activated during AIM2- or NLRC4-inflammasome activation [33]. Thus, this work focused on NLRP3–NEK7 interaction, an upstream event of ASC oligomerization. NLRP3 maintains a closed inactive conformation under normal conditions, but the stimulus-induced interaction of NEK7 and NLRP3 converts the closed NLRP3 into an open structure. Therefore, these conformational changes in NLRP3 are essential for downstream events, including ASC oligomerization, speck formation, and caspase-1 activation [12].

Since the preliminary experiment on NLRP3–NEK7 co-immunoprecipitation in primary mouse macrophage cells was unsuccessful, their interaction was assessed by overexpressing them in HEK293T cells. Indeed, the overexpression of these proteins induced their spontaneous binding, and their interaction was inhibited or dissociated by ODZ treatment (Figure 5). This result indicated that ODZ might suppress NLRP3 inflammasome activation by inhibiting NLRP3–NEK7 interaction.

To further understand which molecule is the target of ODZ, we employed the drug affinity responsive target stability (DARTS) assay, which is based on the reduction in protease degradation of the target protein upon chemical binding [27]. DARTS with cell lysates revealed that the NLRP3 band was more intense in the ODZ-treated lysate compared with vehicle control, indicating that ODZ targets NLRP3. Moreover, ODZ was appropriately placed with the NACHT domain of NLRP3 when it was docked into the NACHT domain of NLRP3 using Autodock Vina (Figure 7). This suggests that ODZ inhibits NLRP3 inflammasome activation through direct interaction with NLRP3.

However, it is important to note that ODZ10117 is a STAT3 inhibitor [21,22], and thus it cannot be ruled out that the inhibition of NLRP3 inflammasome is still a result of the inhibition of STAT3 or a combination of both pathways. Future studies on the precise interplay between the two pathways would provide more definitive conclusions.

To investigate the in vivo relevance of the in vitro activity of ODZ on inflammasome activation, we employed an MSU-induced peritonitis model and LPS-induced sepsis model in mice. Indeed, the treatment of ODZ improved mortality and IL-1β release, suggesting that it can be applicable to inflammasome-associated diseases.

It should be noted that ODZ is a STAT3 inhibitor [21,22] and its inhibitory activity on both STAT3 and NLRP3 inflammasome might be beneficial to cancer treatment. The association of STAT3 with cancer growth has been documented [34,35], and the activation of NLRP3 inflammasome in a cancer microenvironment positively affects cancer growth [36,37,38]. Moreover, the development of Myeloid-derived suppressor cells (MDSCs) is positively regulated by the NLRP3 inflammasome activation [39].

Altogether, this study showed a novel effect of ODZ for the regulation of NLRP3 inflammasome activation in vitro and in vivo, suggesting its potential for use in treating NLRP3-inflammasome-mediated immune disorders and cancer.

## 4. Materials and Methods

### 4.1. Reagents

ODZ was provided by Professor Sang-Kyu Ye (Department of Pharmacology, Seoul National University College of Medicine, Seoul, Republic of Korea) [21]. Penicillin-streptomycin, trypsin-EDTA, Fetal bovine serum, RPMI 1640, DMEM-high glucose pyruvate, and Opti-MEM were purchased from Gibco (Grand Island, NY, USA). LPS (O111:B4), ATP and DMSO were purchased from Sigma-Aldrich (St. Louis, MO, USA). Nigericin, nano-SiO_2_, imiquimod, poly(dA:dT), MSU crystal, Y-VAD-CMK, MCC950, lipofectamine-2000, and mouse IL-1β ELISA kit were purchased from InvitroGen (San Diego, CA, USA). Flagellin was purchased from Adipogen (San Diego, CA, USA). Mouse IL-1β antibody (AF-401-NA) and human IL-1β ELISA kit were purchased from R&D Systems (Minneapolis, MN, USA). Antibodies against mouse caspase-1 (AG-20B-0042-C100), ASC (AG-25B-0006-C100), and NLRP3 (AG-20b-0014-C100) were purchased from Adipogen (San Diego, CA, USA), and mGSDMD (ab209845) was purchased from Abcam (Cambridge, UK). Anti-FLAG antibody (F1804) and β-Actin antibody (SC-4778) were purchased from Sigma-Aldrich (St. Louis, MO, USA) and Santa Cruz biotechnology (Dallas, TX, USA), respectively.

### 4.2. Animals

Female C57BL/6 mice (6–8 weeks old) were purchased from Orient Bio Co. (Seoul, Republic of Korea). The mice were housed in groups of five in a controlled environment (22 ± 2 °C, 55 ± 5% humidity, 12/12 h light/dark cycle) and provided with food and water ad libitum. All experiments were performed under the guidelines of the Konkuk University Animal Care Committee. The experimental protocol was reviewed and approved by the Ethics of Animal Experiments Committee of Konkuk University (#KU21211).

### 4.3. Cell Culture

Bone marrow cells were isolated in the femurs and tibias of C57BL/6 mice, then the bone marrow cells were cultured in RPMI 1640 supplemented with 10% fetal bovine serum (Gibco), 1% penicillin/streptomycin (Gibco), 50 μM β-mercaptoethanol, MEM-NEAA, 1 mM sodium pyruvate, and 30% L929 conditioned medium (LCM) at 37 °C in a 5% CO_2_ atmosphere. The culture medium was added every 3 days, and at day 7, the bone marrow cells were seeded for further experiments.

J774A.1 cells, HEK293T cells and THP-1 cells (Korean cell line bank) were cultured in DMEM or RPMI 1640 supplemented with L-glutamine, 10% fetal bovine serum (Gibco), and 1% penicillin/streptomycin (Gibco) at 37 °C in a 5% CO_2_ atmosphere.

### 4.4. Cell Viability Assay

BMDMs (1.5 × 10^4^/well) were plated on a 96-well plate and incubated overnight. The cells were treated with the indicated concentration of ODZ or vehicle (DMSO) (0.2%) for 6 h, and MTT (3-(4, 5-dimethylthiazolyl-2)-2, 5-diphenyltetrazolium bromide) solution (0.5 mg/mL) was added and further incubated for 2 h. The formazan was dissolved in DMSO, and its optical density (OD) was measured at 550 nm using a Multiskan GO Microplate spectrophotometer (Thermo Fisher Scientific, Waltham, MA, USA). The cell viability was calculated as % = (OD of drug treated sample/OD of non-treated sample) × 100.

### 4.5. Chemical Library and Screening

Approved and investigation chemicals (3110 compounds) were obtained from Korea Chemical Bank (www.chembank.org, accessed on 1 October 2020 ) and from academic organizations in the Republic of Korea. Screening was carried out through the inhibitory effect of chemicals on nigericin-induced IL-1β release in THP-1 cells. PMA-differentiated THP-1 cells (1 × 10^5^ cells/96 well) were pretreated with 50 μM of chemicals, followed by treatment with nigericin (10 μM) for 1 h. Then IL-1β release into the supernatants was measured using ELISA assay.

### 4.6. Inflammasome Activation

BMDMs were seeded on a tissue culture well plate and cultured overnight. The cells were primed with LPS (100 ng/mL) for 3 h and washed twice with sterile PBS. Then, the cells were pre-treated with or without ODZ in Opti-MEM medium for 1 h before stimulating them with various stimuli. For inflammasome activation, the cells were treated with ATP (5 mM) or nigericin (10 μM) for 1 h or silica crystals (150 μg/mL) or imiquimod (30 μg/mL) for 3 h, or transfected with poly (dA:dT) (1 μg/mL) for 2 h or flagellin (1 μg/mL) for 3 h with Lipofectamine-2000.

### 4.7. Enzyme-Linked Immunosorbent Assay (ELISA)

For measurement of IL-1β release, cell culture supernatants or in vivo samples were collected and the cytokine level was analyzed using an ELISA kit (Invitrogen, Waltham, MA, USA) according to the manufacturer’s instructions.

### 4.8. Lactate Dehydrogenase (LDH) Assay

For measurement of cell death, cell culture supernatants were collected and LDH level was analyzed using an LDH assay kit (DoGenBio, Seoul, Republic of Korea) according to the manufacturer’s instructions. The optical density (OD) was measured at 450 nm using a Multiskan GO Microplate spectrophotometer (Thermo Fisher Scientific, Waltham, MA, USA). The LDH release was calculated as % = (OD of each sample/OD of lysis sample) × 100.

### 4.9. Protein Extraction and Immunoblot Analysis

Upon stimulation, the cells were washed with cold PBS and lysed with cold RIPA lysis buffer (iNtRON, Seongnam, Republic of Korea) containing phosphatase and protease inhibitor cocktail (GenDEPOT, P3200, and P3100) on ice for 20 min, centrifuged at 13,000 rpm for 15 min, and the lysates were collected and quantified using a BCA assay kit (Thermo Fisher Scientific, #23227). For immunoblot analysis, cell supernatants and quantified protein samples were boiled with a SDS sample buffer for 5 min, and separated by SDS-PAGE electrophoresis and transferred to a nitrocellulose membrane (Millipore, HATF00010). The membrane was blocked in 5% skim milk for 1 h and incubated with primary antibody overnight at 4 °C. Then, membranes were washed with PBST (0.1% tween-20) for 10 min 3 times and incubated with HRP-conjugated proper secondary antibody for 1 h at room temperature. The target proteins were visualized using an EZ-western Lumi pico kit (DoGenBio, Seoul, Republic of Korea) and detected using an image analyzer, ChemiDoc^TM^ MP (Bio-Rad, Hercules, CA, USA). The densitometric analysis of each Western blot was quantified using Image J 1.53t software. Band densities of each protein were normalized as a ratio to β-actin.

### 4.10. Separation of Triton X-100 Soluble Protein and Insoluble Protein

Cells were lysed with TTNE lysis buffer (1% Triton X-100, 150 mM NaCl, 50 mM Tris, and 2 mM EDTA) containing protease inhibitor cocktail on ice for 20 min and centrifuged at 13,000 rpm for 15 min. The supernatant of this fraction was the soluble protein, and the pellet was the insoluble protein.

### 4.11. Cross-Linking of ASC Oligomers

Cells were lysed with A0 buffer (0.5% Triton X-100, 20 mM HEPES-KOH, pH 7.5, 150 mM KCl, and protease inhibitor cocktail) for 20 min on ice and centrifuged at 6000 rpm for 15 min. The cell pellets were washed with TTNE and PBS at 6000 rpm for 10 min each. Then, they were resuspended with PBS and cross-linked with DSS (2 mM) (Thermo Fisher Scientific, #21555) for 30 min in RT and denatured by 20 mM Tris-HCl for 15 min in RT. The DSS cross-linked pellets were centrifuged at 13,000 rpm for 15 min and dissolved in SDS sample buffer [3].

### 4.12. Immunocytochemistry of ASC Speck

Inflammasome-stimulated cells were washed with cold PBS and fixed with 4% paraformaldehyde for 20 min in 4 °C, permeabilized with acetone for 10 min at −20 °C, and blocked with 10% horse serum for 1 h in RT. Cells were stained with ASC antibody 1:300 (Cell Signaling Technology, #67824S) for 2 h in RT and then Cy3-conjugated rabbit anti-mouse antibody for 1 h at RT in dark. Nuclei were stained with DAPI (Sigma-Aldrich, D9542) [3] and visualized using Nikon fluorescence microscope. ASC speck containing cells were counted using randomly selected fields imaged at 100×magnifications. The merge and cell counting were performed by using Image J 1.53t software.

### 4.13. NEK-NLRP3 Interaction via Co-Immunoprecipitation

The HA-tagged NEK7 gene was amplified from THP-1 cDNA using specific primers that included the HA-tag sequence, and the resulting product was inserted into a plasmid pcDNA3.1 and the sequence accuracy was confirmed through sequencing. The plasmid for DDK-tagged hNLRP3 (RC220952) was purchased from ORIGENE. This process creates a plasmid with the HA-tagged NEK7 gene, which can be used for further experiments. For NEK7-NLRP3 co-immunoprecipitation in a HEK293T overexpression system, HEK293T cells (2.0 × 10^6^) were plated on 100 pi cell culture dish overnight. The pcDNA3.1-HA tagged hNEK7 (5 μg) and pCMV6-Myc-DDK tagged hNLRP3 plasmids (5 μg) were transfected to HEK293T cells by using Lipofectamine-2000 for 24 h. Cells were washed with cold PBS and lysed with IP buffer (1% NP-40, 50 mM Tris-HCl, 150 mM NaCl, 0.5% sodium deoxycholate containing protease inhibitor cocktail) for 20 min on ice and centrifuged at 13,000 rpm for 15 min; then, lysates were precleared with sepharose 4B (Sigma-Aldrich, 4B200) for 2 h. Quantified protein was incubated with flag antibody overnight in 4 °C. The proteins bound to the antibody were pulled down by protein A/G beads 2 h in 4 °C. Protein–antibody complex was extracted by boiling with 2× SDS buffer.

### 4.14. DARTS Assay

Drug affinity responsive target stability (DARTS) assay was performed based on the previous study protocol [40]. LPS-primed J774A.1 were lysed with TTNE buffer, and the lysates were incubated with various doses of ODZ for 30 min. BCA assay was performed and the lysates (2.7 μg/μL) were digested with pronase (1:200 pronase to protein) for 30 min. Additionally, the mixture of lysates was incubated with protease inhibitor cocktail to stop the digestion reaction on ice for 10 min. Additionally, 5× SDS sample buffer was added and boiled for 5 min. Immunoblot analysis was performed.

### 4.15. In Silico Molecular Docking Experiment

To investigate the potential binding of NLRP3 with ODZ, molecular docking was performed using several computer simulation programs. The crystal structure of NLRP3 NACHT domain (PDB ID:7ALV) in RCSB Protein Data Bank was used for protein preparation. The chemical structure of ODZ was obtained using the ChemDraw Professional 20.1.1.125 (PerkinElmer Inc., Waltham, MA, USA). The ligand was confined to the box of ADP binding pocket and employed for ligand–protein docking with Autodock Vina (version 1.2.1). The best output was selected and visualized by Discovery Studio 2022 (BIOVIA; San Diego, CA, USA) software after applying CHARMM force field.

### 4.16. MSU-Induced Peritonitis and LPS-Induced Sepsis

For the MSU-induced peritonitis mouse model, ODZ (5, 10, 20 mg/kg) and MCC950 (20 mg/kg) were administered by intraperitoneal (i.p) injection before 1 h, and after 15 min i.p injection of MSU crystals (1 mg was dissolved in 0.2 mL PBS). After 6 h, mice were sacrificed and peritoneal washing with cold 1 mL PBS was performed. The peritoneal lavage fluid was collected and centrifuged. The supernatant was analyzed by ELISA.

For the LPS-induced sepsis mouse model, ODZ or MCC950 (20 mg/kg) was administered intraperitoneally (i.p) 12 h and 2 h before LPS injection. After 6 h of LPS injection, blood samples were obtained from infraorbital vein blood. The IL-1β release in plasma of blood was analyzed by ELISA and survival rate was assessed up to 7 days after LPS injection.

### 4.17. Statistical Analysis

The results were expressed as the mean ± SEM or SD of at least three independent experiments, each performed in duplicate. Statistical analysis was performed using Dunnet’s post hoc test or log-rank test in Graph Pad Prism 5.0 (San Diego, CA, USA). *p*-values of less than 0.05 were considered statistically significant.

## Figures and Tables

**Figure 1 ijms-24-06079-f001:**
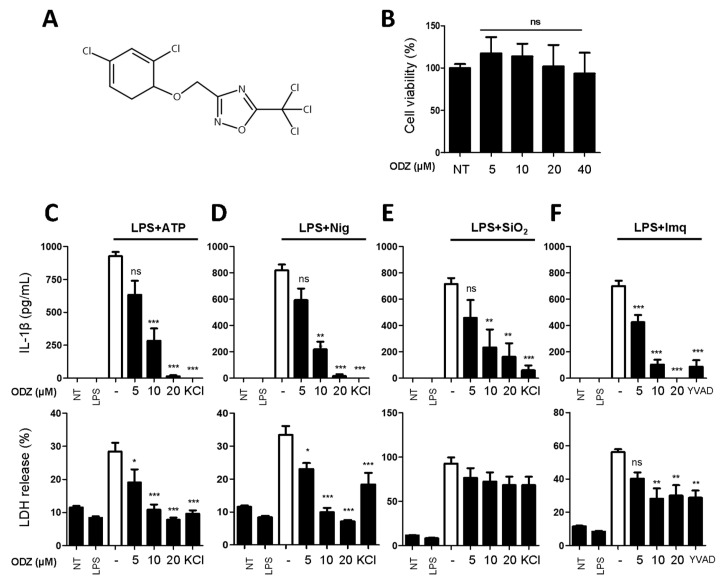
ODZ inhibits IL-1β and LDH release from BMDMs stimulated with various NLRP3 triggers. (**A**) The molecular structure of ODZ; (**B**) BMDMs were treated with the indicated concentration of ODZ for 6 h. Cell viability was measured by MTT assay. Data were expressed as the mean ± SD of three independent experiments performed in triplicate. * *p* < 0.05, ** *p* < 0.01, *** *p* < 0.001, and ns (not significant) compared with non-treated group; (**C**–**F**) LPS-primed BMDMs were pretreated with ODZ dose dependently for 1 h and then stimulated with ATP (5 mM) or nigericin (10 µM) for 1 h, or silica crystals (150 µg/mL) or imiquimod (30 µg/mL) for 3 h. KCl (150 mM) or YVAD (50 µM) was used as positive control. IL-1β release in cell supernatants was measured by ELISA and pyroptosis was measured by LDH assay. Data were expressed as the mean ± SEM of three independent experiments performed in duplicate. * *p* < 0.05, ** *p* < 0.01, and *** *p* < 0.001 compared with LPS-primed BMDMs-treated respective trigger.

**Figure 2 ijms-24-06079-f002:**
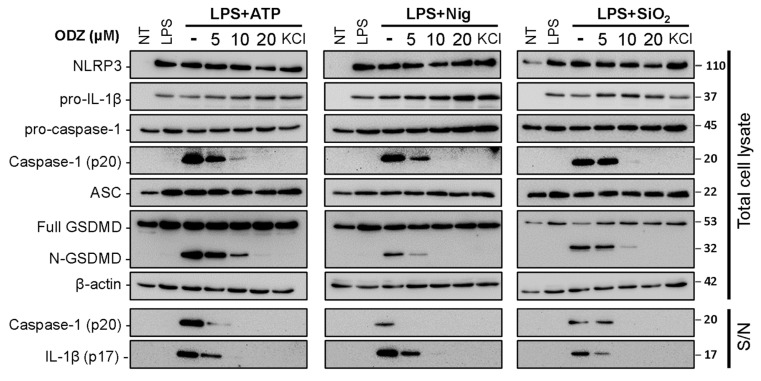
ODZ inhibits IL-1β release and pyroptotic cell death in BMDMs by the suppression of caspase-1 activation. LPS-primed BMDMs were pretreated with the indicated concentration of ODZ for 1 h and then stimulated with ATP (5 mM) or nigericin (10 μM) for 1 h, or silica crystals (150 μg/mL) for 3 h. KCl (150 mM) was used as positive control to inhibit NLRP3-inflammasome activation. The cleaved IL-1β and caspase-1 in culture supernatants (S/N) and the expression level of inflammasome components in quantified cell lysates were analyzed by immunoblot analysis.

**Figure 3 ijms-24-06079-f003:**
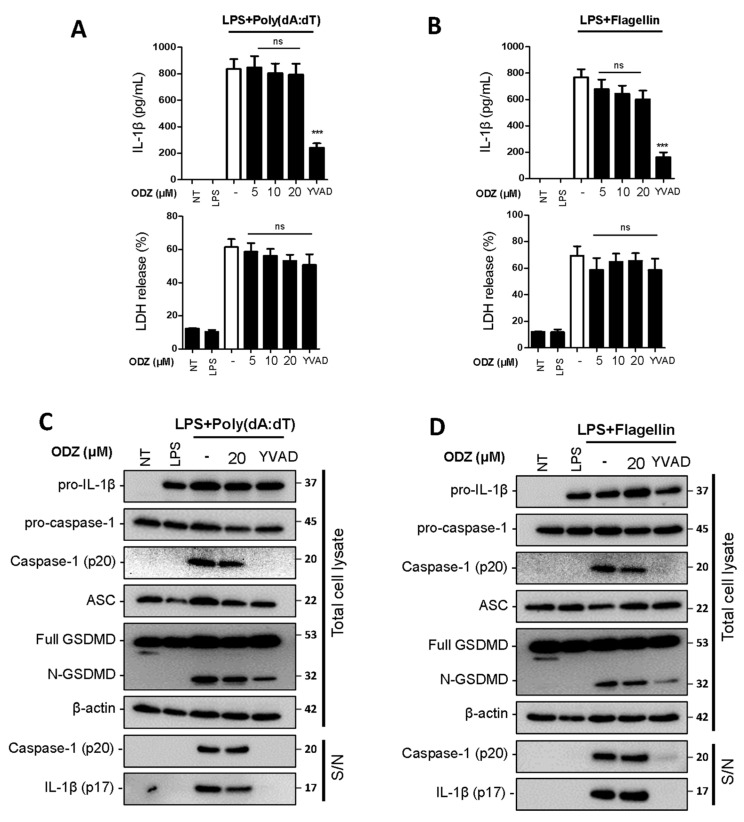
ODZ inhibits neither caspase-1 activation nor IL-1β release induced by AIM2- or NLRC4- inflammasome activation. (**A**–**D**) LPS-primed BMDMs were pretreated with the indicated concentration of ODZ for 1 h and then transfected with poly(dA:dT) (1 μg/mL) for 2 h or flagellin (1 μg/mL) for 3 h. YVAD (50 µM) was used as positive control to inhibit the IL-1β release. (**A**,**B**) IL-1β release in cell culture supernatants was measured by ELISA and pyroptosis was measured by LDH assay; (**C**,**D**) Inflammasome-related target proteins in culture supernatants (S/N) and quantified cell lysates were analyzed by immunoblot analysis. Data were expressed as the mean ± SEM of three independent experiments performed in duplicate. *** *p* < 0.001 and ns (not significant) compared with LPS-primed BMDMs-treated respective trigger.

**Figure 4 ijms-24-06079-f004:**
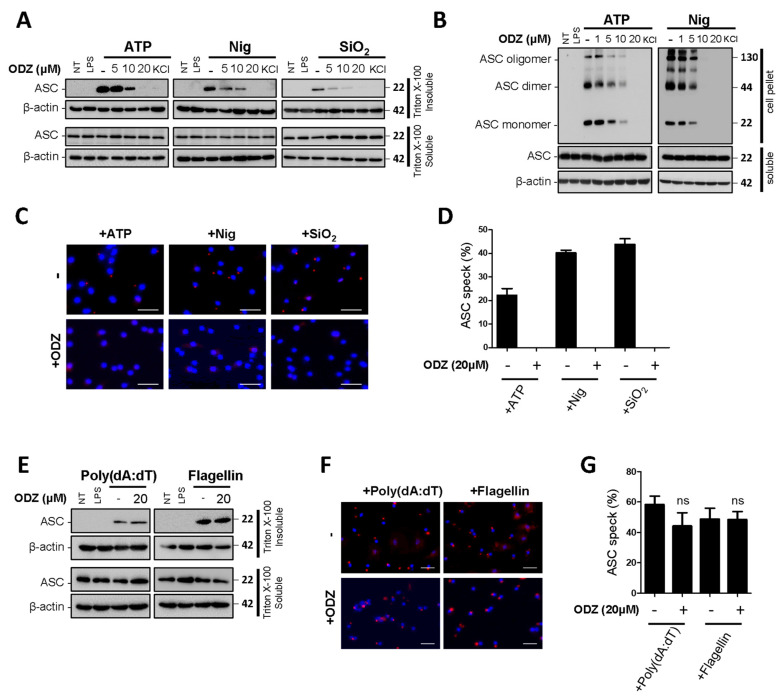
ODZ inhibits ASC redistribution, its oligomerization, and speck formation. (**A**–**D**) LPS-primed BMDMs were pretreated with the indicated concentration of ODZ for 1 h and then stimulated with ATP (5 mM) or nigericin (10 µM) for 1 h or silica crystals (150 µg/mL) for 3 h. KCl (150 mM) was used as a positive control; (**E**–**G**) LPS-primed BMDMs were pretreated with the indicated concentration of ODZ for 1 h and then transfected with poly(dA:dT) (1 μg/mL) for 2 h or flagellin (1 μg/mL) for 3 h. (**A**,**E**) ASC translocation in Triton X-100 insoluble fraction was analyzed by immunoblot analysis; (**B**) DSS cross-linked ASC oligomerization was analyzed by immunoblot analysis; (**C**,**F**) ASC speck formation was analyzed by immunocytochemistry. Cells were stained for ASC (red), DAPI (blue); (scale bars 20 µm) (**D**,**G**) The percent of ASC speck formation was calculated as the ratio of the number of ASC positive cells to the number of DAPI positive cells. Cells were randomly selected at least three fields and counted more than 100 cells for each field. Data were expressed as the means ± SDs of three independent experiments; ns (not significant).

**Figure 5 ijms-24-06079-f005:**
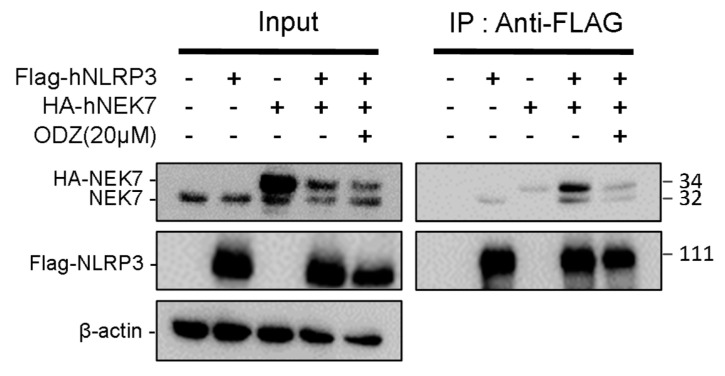
ODZ inhibits the interaction of NLRP3 and NEK7. FLAG-NLRP3 and HA-NEK7 were overexpressed in HEK293T cells, and their interaction was analyzed by immunoprecipitation and immunoblot analysis.

**Figure 6 ijms-24-06079-f006:**
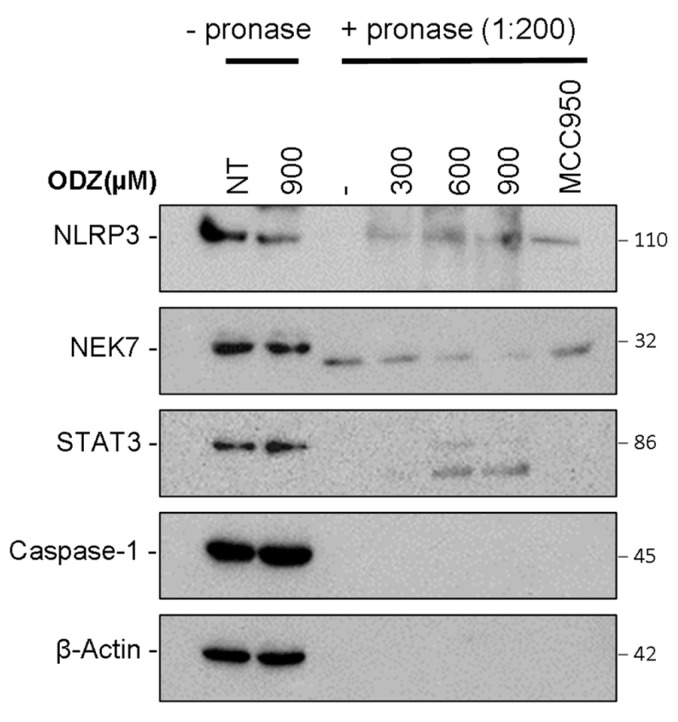
Drug affinity responsive target stability (DARTS) showed that ODZ might bind to NLRP3. LPS-primed J774A.1 lysates were incubated with the indicated concentration of ODZ for 30 min and the lysates were treated with pronase (0.135 μg/mL) for 30 min. The immunoblot analysis of NLRP3, STAT3, NEK7, caspase-1, and β-actin was performed.

**Figure 7 ijms-24-06079-f007:**
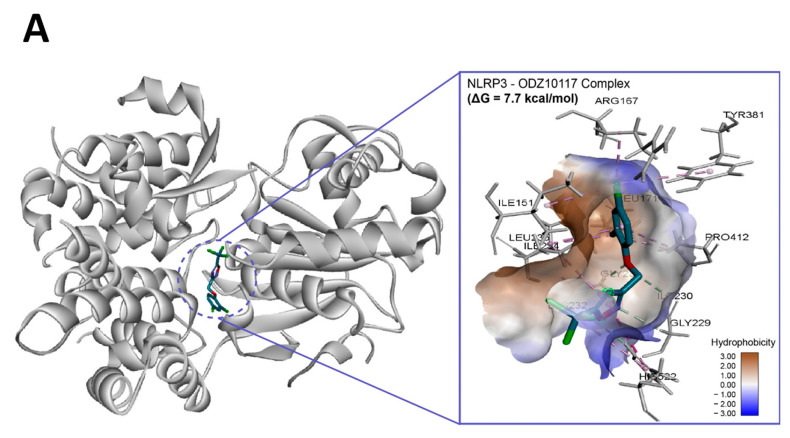
In silico binding analysis of ODZ with NLRP3 that was executed by Autodock Vina (version 1.2.1) (**A**) Binding pose of ODZ docked into the NACHT domain; (**B**) Ligand interaction diagram of ODZ binding pose in NLRP3. The legend and surfaces of the model were plotted using discovery studio 2022 (BIOVIA; CA, USA) and hydrophobicity is shown in blue to brown with values ranging from −3 (blue) to +3 (brown).

**Figure 8 ijms-24-06079-f008:**
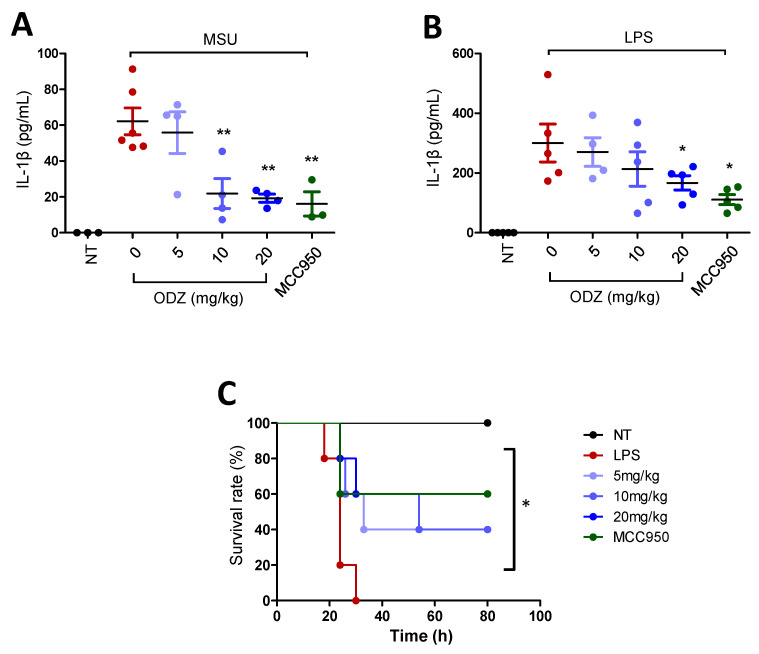
In vivo relevance of ODZ on peritonitis and sepsis model. (**A**) The MSU-induced peritonitis model. An indicated amount of ODZ or MCC950 (20 mg/kg) was intraperitoneally injected and followed by the injection of MSU crystals (50 mg/kg). After 6 h, peritoneal lavage fluid was obtained and the level of IL-1β was assessed by ELISA assay. NT group (*n* = 3), MSU group (*n* = 6), MSU + ODZ group (*n* = 4), MSU + MCC950 group (*n* = 3); (**B**–**C**) The LPS-induced sepsis model. ODZ or MCC950 (20 mg/kg) were intraperitoneally injected 12 h and 2 h before LPS (20 mg/kg) injection (*n* = 5). After 6 h, plasma of blood was collected for ELISA assay and survival rate was monitored daily up to 7 days. (**B**) IL-1β release. Statistical analysis was calculated by *t*-test; (**C**) Survival rates was analyzed by log-rank test. Data were expressed as the mean ± SD; * *p* < 0.05 and ** *p* < 0.01 compared with LPS or MSU injected group.

## Data Availability

Not applicable.

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
