# Peer review of "Novel Activity of ODZ10117, a STAT3 Inhibitor, for Regulation of NLRP3 Inflammasome Activation"

_ijms, 2023, doi:10.3390/ijms24076079_

Round 1
Reviewer 1 Report
In this study, the authors investigated the inhibitory activity of a chemical compound, namely ODZ10117 (ODZ), on NLRP3 inflammasome activation.
The authors reported that ODZ inhibited the cleavage of caspase-1, GSDMD, and pro-IL-1β, but it did not affect the expression of NLRP3, ASC, GSDMD, pro-caspase-1, and pro-IL-1β.
Co-immunoprecipitation, DARTS studies and In silico binding analysis, suggested that ODZ inhibits NLRP3 inflammasome activation by impeding the interaction between NEK7 and NLRP3, through binding with NLRP3.
The study is interesting and it has been conducted by using also sophisticated methodological approaches
Regarding the interpretation of western blot results, the authors should represent the modulation of the investigated proteins (with respect to the internal loading), by using histograms or the numbers above each blot.
The authors often emphasized the anti-cancer effects of ODZ, pointing out that this effect could also result from the inhibitory activity on NLRP3 inflammasome, as well as on STAT3 (lines 56-57; 271-279). I agree since the pro-inflammatory tumor microenvironmental plays a crucial role in the progression of cancer progression and aggression. To further demonstrate that ODZ counteracts the NLRP3-dependent inflammation, the authors tested the biological efficacy of ODZ in an LPS-induced sepsis model. I think that it should be more coherent and forward-looking, to use an in vivo cancer model, through which the authors could demonstrate the potential of ODZ. Anyway, it is a personal comment.
Author Response
Response to Reviewer #1:
In this study, the authors investigated the inhibitory activity of a chemical compound, namely ODZ10117 (ODZ), on NLRP3 inflammasome activation.
The authors reported that ODZ inhibited the cleavage of caspase-1, GSDMD, and pro-IL-1β, but it did not affect the expression of NLRP3, ASC, GSDMD, pro-caspase-1, and pro-IL-1β.
Co-immunoprecipitation, DARTS studies and In silico binding analysis, suggested that ODZ inhibits NLRP3 inflammasome activation by impeding the interaction between NEK7 and NLRP3, through binding with NLRP3.
The study is interesting and it has been conducted by using also sophisticated methodological approaches
Response: Thank you for taking to review our paper. We greatly appreciate your feedback and suggestions, which have helped us improve the quality of our work.
Regarding the interpretation of western blot results, the authors should represent the modulation of the investigated proteins (with respect to the internal loading), by using histograms or the numbers above each blot.
Response: As per your suggestion, we have included the histograms of the data of western blotting in supplementary figures. This was done because including all graphs in the main figure would have made it excessively long. We believe that this addition will provide a clearer understanding of our findings.
The authors often emphasized the anti-cancer effects of ODZ, pointing out that this effect could also result from the inhibitory activity on NLRP3 inflammasome, as well as on STAT3 (lines 56-57; 271-279). I agree since the pro-inflammatory tumor microenvironmental plays a crucial role in the progression of cancer progression and aggression. To further demonstrate that ODZ counteracts the NLRP3-dependent inflammation, the authors tested the biological efficacy of ODZ in an LPS-induced sepsis model. I think that it should be more coherent and forward-looking, to use an in vivo cancer model, through which the authors could demonstrate the potential of ODZ. Anyway, it is a personal comment.
Response: Thank you for your thoughtful comments and we agree with the reviewer’s opinion. Nevertheless, the challenge with ODZ is that it impacts both STAT3 activation and NLRP3 inflammasome activation, making it intricate to distinguish which mechanism specifically contributed to its anti-cancer prosperity in an in vivo model. Therefore, we utilized a inflammasome-related model to demonstrate its in vivo inhibitory activity on the inflammasome.
Above all, we would like to express our genuine gratitude for the time and effort that our reviewers have put into reviewing our manuscript and providing insightful comments.

Author Response
" please see the attachment"

Reviewer 3 Report
Kang et al have addressed the role of the STAT3 inhibitor ODZ10117 in regulation of NLRP3 activation. Most of the experiments are well performed and clearly presented. However the authors have not performed crucial experiments looking at the involvement or ruling out the role of STAT3 per se in NLRP3 activation which is important since they are exploring activity of a STAT3 inhibitor. My comments are as follows:
1. STAT3 has been shown to modulate NLRP3 inflammasome activation.
https://onlinelibrary.wiley.com/doi/abs/10.1002/syn.22221
https://acrabstracts.org/abstract/evidence-that-stat3-controls-nlrp3-inflammasome-dependent-release-of-il-1%CE%B2-and-pyronecrosis-through-regulation-of-mitochondrial-activity/
2. The authors should independently assess the role of STAT3 per se in their experiments. It is highly likely that the effect of ODZ is via STAT3. This could be done by using (i) Other STAT3 inhibitors (ii) siRNA mediated ablation of STAT3. In these cells the role of ODZ should be assessed in regulation of NLRP3 activation. If ODZ treatment attenuates NLRP3 activation in these cells then it is believable that ODZ inhibits NLRP3 independent of STAT3.
3. Other STAT3 inhibitors such as BP-1-102, WP1066 amongst many should be tested. This will inform if the activity is unique to ODZ. Also use of other inhibitors along with ODZ will reveal effect of STAT3 in NLRP3 activation.
4. The authors have not shown the screen data which is the basis of the study. (Line 69)
5. Figure 1 - Why does ODZ or KCl not inhibit LDH release in SiO2 treated cells (1E)? Y axis of 1D is mislabelled.
6. Figure 3 - Why does YVAD not inhibit LDH release?
7. Figure 5 - Legend is totally missing. Therefore one cannot interpret the data.
8. The mode of action of Pronase should be explained.
9. Figure 8 - In vivo experiments should be done with either mice with conditional deficiency of STAT3 or administering mice other STAT3 inhibitors which inhibit STAT3 but not NLRP3. In these conditions mice should be treated with NLRP3 activators in presence or absence of ODZ.
Author Response
"please see the attachment"

Round 2
Reviewer 2 Report
I would like to thank the authors for thoroughly addressing my comments. I recommend the manuscript be accepted in present form.